# Adolescent Young Carers Who Provide Help and Support to Friends

**DOI:** 10.3390/healthcare11212876

**Published:** 2023-11-01

**Authors:** Rosita Brolin, Elizabeth Hanson, Lennart Magnusson, Feylyn Lewis, Tom Parkhouse, Valentina Hlebec, Sara Santini, Renske Hoefman, Agnes Leu, Saul Becker

**Affiliations:** 1Department Health and Caring Sciences, Linnaeus University, SE-39182 Kalmar, Sweden; lennart.magnusson@lnu.se; 2The Swedish Family Care Competence Centre, Strömgatan 13, SE-39232 Kalmar, Sweden; 3School of Nursing 179, Vanderbilt University School of Nursing, 461 21st Avenue South, Nashville, TN 37240, USA; feylyn.m.lewis@vanderbilt.edu; 4School of Psychology, University of Sussex, Falmer, Brighton BN1 9QQ, UK; tp93@sussex.ac.uk; 5Faculty of Social Sciences, University of Ljubljana, 1000 Ljubljana, Slovenia; valentina.hlebec@fdv.uni-lj.si; 6Centre for Socio-Economic Research on Ageing, IRCCS INRCA-National Institute of Health and Science on Ageing, 60124 Ancona, Italy; s.santini2@inrca.it; 7The Netherlands Institute for Social Research (SCP), Postbus 16164, 2500 BD The Hague, The Netherlands; r.hoefman@scp.nl; 8Institute for Biomedical Ethics, Science and Medical Faculty, University of Basel, 4001 Basel, Switzerland; agnes.leu@unibas.ch; 9School of Education and Social Work, University of Sussex, Falmer, Brighton BN1 9QQ, UK; s.becker@mmu.ac.uk; 10Faculty of Health and Education, Manchester Metropolitan University, Manchester M15 6BX, UK

**Keywords:** young friend carer, young carer, school outcomes, health, well-being

## Abstract

Prior studies emphasize the value of friends’ support for children/adolescents who have a disability or suffer from mental ill-health or a long-term illness. However, few studies have explored how a caring role affects those young friend carers themselves. This paper addresses a gap in the research by focusing on this hitherto neglected group of young carers to explore the impact of providing care to friends. An online survey was employed for a cross-national study conducted in 2018–2019 in Sweden, Italy, Slovenia, the Netherlands, Switzerland, and the United Kingdom to examine the incidence of adolescent young friend carers, the extent of care they provide, and their self-reported health, well-being, and school situation. The survey was completed by 7146 adolescents, aged 15–17, and 1121 of them provided care to a friend with a health-related condition, most frequently mental ill-health. They carried out high levels of caring activities, and a quarter of them also provided care to a family member. They experienced both positive and negative aspects of caring. Nevertheless, in comparison with adolescents who provided care to family members, they reported more health problems, with a dominance of mental ill-health, and they received lower levels of support. Since adolescent friends play a valuable role for young people with health-related conditions, especially mental ill-health, it is important to find ways of optimizing their caring experiences in order that those adolescents who choose to care for a friend can do so without it having a negative impact on their own mental health, well-being, and life situation.

## 1. Introduction

Childhood is commonly seen as a stage of life free from extensive responsibilities, where adults provide care and protection, while children are expected to spend their time playing, going to school, and hanging out with friends. However, many children and adolescents show considerable civic engagement and capability, as evidenced by the recent Black Lives Matter Youth Vanguard [1], Teens 4 Equality [2], Malala Education Foundation [3], Earth Guardians [4], and Fridays for Future [5], to name but a few examples of young people’s engagement on a macro level.

At the micro level, many children and adolescents take responsibility in their closest social network by providing care and/or support to a family member or a friend with a long-term illness, disability, mental ill-health, or addiction. These children and adolescents are defined in the literature as young carers (YCs) [6] (p. 378). The awareness of YCs in Europe is relatively low [7,8] but the proportion of YCs has been estimated to be about 5–10% of the youth population in Europe [9].

The original definition of YCs is “children and young people under 18 who provide or intend to provide care, assistance, or support to another family member. They carry out, often on a regular basis, significant or substantial caring tasks and assume a level of responsibility which would usually be associated with an adult” [6] (p. 378). This definition only includes young family carers. However, we now know that there are also young people who provide care to individuals who are not family members, but who are still close to them, usually a close friend [9]. In this paper, they are called “young friend carers”. The awareness of this specific group of YCs is likely to be very low as they are not mentioned in the original definition (which is over 20 years old), nor in legislation [7], nor more generally in innovative practices/service provision for young carers.

Regardless of people’s age and life situation, friends play an important role in one’s life and have proven to be especially important when people are faced with challenges and go through difficulties or major changes in life [10,11,12]. During the COVID-19 pandemic, family members and friends often had to step in and provide care and support to a greater extent than previously [13,14].

During the transition to adulthood, adolescents may begin to confide in and seek support from friends rather than from parents or other adults [15,16]. High-quality friendship can be an important buffer against adolescent adjustment problems in school [17]. Furthermore, friends’ support is important for children/adolescents who have a disability or suffer from long-term illness or mental ill-health [18,19,20,21,22,23].

Mental health disorders are common among children and adolescents [24,25] and have been on the increase in recent decades [26,27,28,29,30,31,32]. Mental ill-health affects not only the affected person, but also family members and friends who provide care for them [33,34,35], with psychological distress [35], fear [33], elevated burden, grief, mental health problems, and interpersonal strain [34]. Adolescents’ over-involvement in friends’ problems can have a negative impact on their own well-being [36]. In an Australian study, more than half of the adolescent sample stated that they would provide social support if their friend suffered from mental ill-health. This was more common among girls than among boys [37]. To the best of our knowledge, there are no systematic studies that focus entirely on young friend carers. However, there are several studies that focus on YCs who provide care or support to a family member or a friend. Being a YC has been proven to be a risk factor for mental health and well-being [38,39], health inequalities during young people’s life cycle [40,41], and social exclusion, with higher levels of absence and drop-outs from education and lower employment rates [40,42].

On the other hand, a caring role can lead to personal growth and competence, with feelings of inner strength and self-confidence; thus, it can positively affect mental well-being [43]. This balance between negative and positive caring experiences is illustrated in a small-scale study of five American adolescents who provided support to friends with mental ill-health. They expressed that their caring role made them feel honored, important, and capable, while, at the same time, there were challenging aspects, such as being fearful, maintaining vigilance, keeping secrets, setting boundaries, seeking knowledge, and making decisions about involving others or not [44]. Recent research indicated that adolescents who provide care to grandparents can experience more positive caregiving outcomes than adolescents who provide care to other close persons [45].

American family/friend young adult cancer caregivers were found to seek support through social media [46]. Swedish researchers found that providing care to a family member or a friend with mental illness made YCs’ daily lives unpredictable [47]. In the same study, it was highlighted that very few of them received professional support, while their main support came from parents, siblings, and friends [48]. Young family carers experienced a lower level of support, while young friend carers experienced a lower positive value of caring [49].

Previous studies have shown the importance of friends’ care and support for young people with mental ill-health, physical illness, injury, or disability, and that the need for friends’ care and support increases during crises. At the same time, there is an overall lack of empirical studies with a focus on young friend carers, which demonstrates the need for research in this area. Previous empirical studies have revealed positive, as well as negative effects and risk factors of being a YC. Most of these studies focused on young Family Carers. Some studies focused on YCs who provide care to a family member or a friend, but only one study made comparisons between the two different groups of YCs [49]. Finally, only one small-scale study focused entirely on young friend carers [44]. The goal of this study is to contribute to knowledge to fill this research gap by focusing on adolescent young friend carers and adolescent young family and friend carers who provide care, help and/or support to both a family member and a friend.

The present study forms part of the first major cross-national empirical study in the larger EU Horizon 2020-funded research and innovation project: psychosocial support for promoting mental health and well-being among adolescent young carers in Europe—“ME-WE” [50]. According to the United Nations Convention on the Rights of the Child (UNCRC) [51], article 12, all children have the right to express their views and to be listened to. The European Commission recently stipulated that no policy regarding children should be designed without their voices being heard [52]. In accordance with the UNCRC and the EU Strategy, the ME-WE project assumed that each adolescent is an expert on their own unique life situation. Thus, the project started with an online survey, targeted at adolescents, with the aim of gaining insights into the profiles, caring activities, needs and preferences of adolescent young carers (AYCs) in six European countries—Sweden, Italy, Slovenia, the Netherlands, Switzerland, and the United Kingdom. The legal situation in the six partner countries is varied, from the United Kingdom, with most targeted legislation regarding young carers, to Sweden, with legal provision in the Health Care Act, and further to a lack of national legislations in the Netherlands, Switzerland, Italy, and Slovenia [7]. The survey was completed by 7146 adolescents (15–17 years), of which 2099 were identified as AYCs, caring for family members, friends, or other close persons [53].

The aim of this study was to gain increased knowledge about adolescent young friend carers and adolescent young family and friend carers in six European countries, the extent of the care they provide, how they perceive that caregiving affects their health, well-being, and school outcomes, and what kind of support they receive. A further aim was to compare their situation with the situation of adolescents who provide care to family members only and those who do not provide care to anyone.

## 2. Materials and Methods

The questionnaire, which was especially created for the first online survey in the ME-WE project [53], first included demographic questions about age, gender (including gender identity), place of residence, nationality/citizenship, and family composition. There then followed three validated instruments with high reliability: The Multidimensional Assessment of Caring Activities (MACA-YC18) [54]; Positive and Negative Outcomes of Caring (PANOC-YC20) [54]; Kidscreen-10 [55]. In addition, the questionnaire included a section on education, employment, self-rated general health status, and current access to formal and informal support. The friend carers were identified by their responses to four questions. The first question was about having friends with a health-related condition. The second was a multiple-choice question on these friends’ type of health-related conditions: physical disabilities (e.g., caused by frailty, accident, injury, or illness); mental illness (e.g., depression or anxiety); cognitive impairments (e.g., autism, learning disorders, traumatic brain injury, Downs syndrome, dementia, or Alzheimer’s); addiction (e.g., drugs or alcohol); or other health-related conditions (describe with own words). The third was a multi-choice question about their relation to these persons (girlfriend, boyfriend, partner, friend, colleague, neighbor, ex-girlfriend, ex-boyfriend, ex-partner, cohabitant, or other person). Finally, they were asked if they look after, help, or support any of these friends. Similar questions were also asked corresponding with adolescents’ family members, to identify family carers.

The survey was published on a web page where the users were able to choose their language before receiving a brief description of the study and a link to the survey. To guarantee participants’ anonymity and privacy, the 1 ka platform was used. The online survey was designed to be completed on different types of electronic devices. Due to the limited availability of electronic devices, a paper questionnaire was used in Italy and in exceptional cases in the other five countries. The paper data were then transferred to the online platform by country partner teams and checked for data entry accuracy. In total, completion of the questionnaire took about 30 min maximum.

There were two inclusion criteria at data collection: (1) being aged between 15–17 years and (2) being available to fill in the questionnaire. All country partners’ recruitment strategies involved the targeting of schools. To reduce the risk of sampling bias, a multistage facility sample was adopted: first, regional differentiation within the countries, ensuring participants from urbanized, from less-urbanized and rural areas; second, various recruitment channels: schools, care organizations, care recipients’ interest groups, and municipalities [53]. Due to recruitment challenges, it was not always possible to adhere to the recruitment strategy. In some countries, it was difficult to reach rural areas. In Switzerland and Slovenia, recruitment took place in vocational schools, and in Switzerland, only in the German-speaking part. Italy recruited mainly in high schools of two regions, and Sweden recruited mainly in schools in one county. The Netherlands recruited in schools, care support centres, patient and carer organizations, and social media channels, while the United Kingdom mainly recruited through formal support organizations for young carers, young carers’ festivals, social media, and a small number of schools [53].

Data were collected in two periods: April 2018–December 2018 (all six countries); and January 2019–July 2019 (Sweden, Switzerland, The Netherlands, and United Kingdom only). In Sweden, Switzerland, Italy, Slovenia, and parts of the Netherlands and the United Kingdom, the data collection was carried out during class at school, and all pupils in the respective classes were invited to fill in the questionnaire, while some respondents in the Netherlands and United Kingdom completed the online questionnaire during their leisure time.

Collected data were analyzed using IBM SPSS Statistics (version 25.0, Armonk, NY, USA: IBM Corp.). Descriptive data, including frequency, mean, and standard deviation, were used for the demographic parts. Descriptive and inferential statistics, including independent-samples *t*-tests, paired-samples *t*-tests, and one-way ANOVA, were used to compare findings between different groups.

All country partners secured formal ethical approval from relevant ethics committees or detailed ethics opinions (Switzerland) in their respective countries in accordance with national legislation in April 2018. Respondents were recruited on a voluntary basis according to international declarations, regulations, and guidance documents [56,57,58]. To ensure anonymity, no registration was required for participants to access the survey.

The online survey started with an information page and consent form, written in clear, easy to understand language, appropriate to the participants’ age. The information included all relevant aspects of the research protocol, foreseen benefits, possible risks of participation, and stated that participation was anonymous and that participants had the right to withdraw at any time without consequences. Country partners also added their country-specific referral mechanisms with respect to external education, care, and support professionals in case of need. Once they had given their informed consent, they could proceed to the questionnaire. Those participants who completed the survey on paper received an information sheet and consent form on paper, and a member of the research team collected their informed consent on paper before giving them the questionnaire. Where necessary due to national legislation, partners also secured informed consent by the participants’ legal guardians. Each country partner followed the General Data Protection Regulations [59].

To ensure appropriate security and data protection, a commercial provider was used to host the survey. No data were collected that identified any individual respondent, and each response was designated an identity number. Collected data were stored securely and encrypted to an appropriate standard under the University’s information security policies (https://www.sussex.ac.uk/infosec/policies accessed on 21 August 2023); paper documents are stored in a locked filing cabinet. Only authorized members of the research team have access to the data.

## 3. Results

### 3.1. Participant Characteristics

The survey was completed by 7146 adolescents, aged 15–17 years, of which 60.1% were female, 38.3% male, and 1.6% transgender, non-binary, or other gender. Most respondents (64.5%) lived in urban areas, and the majority (90.6%) were born in the country where the survey took place.

Out of the 7146 respondents, 2055 (29.2%) reported that they had at least one friend or other close person (not family member) with a health-related condition, and 1121 (56.9%) of them provided care, support, or assistance to their friend(s) and were thus identified as friend carers, with the highest proportion in Sweden, and the lowest in the Netherlands and Switzerland (Table 1).

The total number of identified AYCs was 2099 (29.4% of the total sample). Out of these 2099 AYCs, 655 (31.2%) were identified as friend carers, 978 (46.6%) as family carers, and 466 (22.2%) provided care to both family member/s and friend/s and were thus identified as family and friend carers. The majority of respondents in all groups were female (Table 2).

### 3.2. Care Recipients

The 655 participating friend carers provided care to a total of 838 friends or other close persons who were not family members. Most of their care recipients were close friends (N = 511) but girlfriends/boyfriends/partners (N = 123), ex-girlfriends/ex-boyfriends/ex-partners (N = 79), colleagues (N = 40), neighbors (N = 28); cohabitants/flatmates (N = 6), and other close persons (e.g., cousins, nephews, friends’ siblings) (N = 121), were also included, and 7% of the friend carers lived with their care recipient. The 466 family and friend carers provided care to 579 family members and 652 friends.

The AYCs were also asked about what kind of health-related condition their friend(s) and/or family member(s) have. The most commonly reported health-related condition was mental ill-health, which was much more commonly reported by friend carers and family and friend carers compared to the family carers group, while physical disabilities were more commonly reported by family carers (Table 3). Addiction, cognitive impairments, and other health conditions were also mentioned, and some friends and family members had more than one health-related condition.

### 3.3. The Extent and Nature of Caring Tasks

The amount of caring activities was measured with MACA-YC18 and analyzed with one-way ANOVA, with higher scores indicating more caring activities. Compared to their non-carer peers (M = 8.81, SD = 4.57), AYCs performed greater amounts of caring activities (M = 12.57, SD = 5.64), t (3210.93) = 26.73, *p* < 0.001, d = 0.73, with a mean value of 11.13 (SD 5.21) for friend carers, 12.70 (SD 5.56) for family carers, and the highest mean value 14.29 (SD 5.85) for family and friend carers, df = 3, F = 324.397, *p* > 0.001. Friend carers’ self-reported caring activities were similar to those of family carers, with a clear dominance of domestic and household tasks and emotional care (Table 4).

The positive and negative effects of the caring role were assessed with the PANOC-20 positive and negative scale, with scores below 12 on the positive scale and/or scores above 8 on the negative scale indicating potential concern. The friend carers’ results on the positive scale (M = 12.93) and the negative scale (M = 4.93) were similar to those of family carers (M = 12.80, and M = 4.77). The lowest scores on the positive scale (M = 12.49) and the highest scores on the negative scale (M = 7.07) were found among the family and friend carers, who also had the highest proportion of scores below 12 on the positive scale and the highest proportion of scores above 8 on the negative scale (Table 5).

The majority of AYCs in all three groups felt good about caring. They reported that caring led them to learn useful things, that caring had positive impacts on their self-view, their relations to families, and their ability to cope with problems. On the other hand, a majority in all three groups felt stressed about caring and could not stop thinking about their caring responsibilities. Furthermore, about half of the family and friend carers felt very lonely and that they could not cope (Table 6).

### 3.4. The AYCs’ Health Condition and School Situation

The adolescents rated their general health status from 1 = ”Excellent” to 5 = ”Poor” with higher scores indicating poorer health. The one-way ANOVA results showed a poorer self-rated general health among friend carers (M = 2.94, SD = 1.11) compared to their non-carer peers (M = 2.41, SD = 1.03), as well as compared to family carers (M = 2.74, SD = 1.06), while family and friend carers reported the poorest general health (M = 3.37, SD = 1.11), df = 3, F = 160.309, *p* < 0.001. The most common health-related condition was mental health problems, with the highest incidence among family and friend carers, followed by friend carers and family carers, with the lowest incidence among non-carers (Table 7).

The adolescents’ health-related quality of life was measured with KIDSCREEN-10, in which a total score of 50 indicates an extremely high level of well-being. In comparison to their non-carer peers (M = 36.72, SD = 6.53), the one-way ANOVA results showed lower levels of health-related quality of life for friend carers (M = 33.53, SD = 6.98) as well as for family carers (M = 33.97, SD = 7.28), while the lowest levels were found among family and friend carers (M = 30.43, SD = 7.55), df = 3, F = 167.749, *p* < 0.001.

Some AYCs in all groups reported that the caring role had a negative impact on their school situation. This was most common among family and friend carers, followed by family carers. In some cases, AYCs stated that due to caring, they had thoughts of hurting themselves, the care recipient, or someone else. Thoughts of hurting oneself were more than twice as common among family and friend carers, in comparison to friend carers and family carers, while thoughts of hurting the care recipient were most common among family carers (Table 8).

### 3.5. AYCs’ Access to Support

AYCs were also asked about their access to formal and informal support. Consistent for all three groups was that the support they received mainly came from their friends. The friend carers reported the least access to formal as well as informal support (Table 9).

## 4. Discussion

The aim of this study was to gain increased knowledge about adolescent young friend carers and adolescent young family and friend carers, the extent of the care they provide, their self-reported health problems, well-being, and school outcomes, as experienced by them, and what kind of support they receive. A further aim was to compare their situation with the situation of adolescent young family carers and non-carers.

The results showed that 29.2% of all respondents had at least one friend with a health-related condition. More than half of those respondents (56.9%) provided care to their friend, with the highest proportion in Sweden (66.2%) and the lowest in Switzerland (45.5%) and the Netherlands (47.4%), respectively. Further studies are needed to determine the causes for this variation.

The most common health-related condition among the care-receiving friends was mental ill-health, which can be seen as a consequence of the high and increasing levels of mental ill-health among young people [24,25,26,27,28,29,30,31,32]. However, it is important to be aware that no conclusions can be drawn regarding the proportion of professional diagnoses, as the care recipients’ health-related conditions were self-reported by the AYCs.

Adolescents often prefer to confide in and seek support from friends rather than from adults [15,16], especially when it comes to mental health issues, because of stigma and lack of trust [23]. Thus, increasing numbers of young people with mental ill-health may mean increasing numbers of young friend carers.

More than one out of five (22.2%) AYCs were identified as family and friend carers, which means that in addition to providing care to at least one friend, they also provided care to at least one family member. This study provides, for the first time, comprehensive insight into adolescent young friend carers’ and adolescent young family and friend carers’ caring situations and the consequences of their caring roles. Friend carers performed caring activities similar to those of family carers, and they provided care at almost as high levels as family carers, while family and friend carers provided care at a much higher level.

The impacts of a friend-caring role were similar to those of a family-caring role, with both positive and negative PANOC scores, and a majority in all three groups felt good about caring. The positive impacts of caring, which were observed in all three groups, confirms previous empirical study findings in the area of young carers [43]. The 655 friend carers’ and 466 family and friend carers’ commitment and capacity are clearly shown by the fact that they provided care and support to a total of 2069 people. The downside of this commitment is that it can negatively affect their health, well-being, school situation, and school results, with increased risks of social isolation and a potentially enduring negative impact. A majority felt stressed about their caring role, and the PANOC results indicated potential concern for 15.3% of the friend carers who scored above 8 on the negative scale, as well as for 24.3% of the friend carers who scored below 12 on the positive scale. These figures were higher for family carers (16.6% and 29.4%), contradicting a previous study [49], and highest among family and friend carers (33.0% and 33.3%).

Friend carers reported lower health-related quality of life than non-carers and lower general health than both family carers and non-carers, with a higher prevalence of self-reported mental ill-health. These results indicate that young friend carers need attention to the same extent as young family carers. Specific attention should be paid to young family and friend carers who scored lowest on the PANOC positive and highest on the negative scale and reported the poorest general health and highest prevalence of mental health problems. These results provide a picture of the complexity of their situation, with care both within and outside the family, which likely means fewer opportunities to take a pause from their caring role.

The high prevalence of mental health problems among friend carers and family and friend carers could be because most of their care recipients suffering from mental ill-health, which has been shown to negatively affect mental health and well-being among informal carers of all ages [33,34,35]. For some AYCs, the caring role led to difficulties in school, which was most common among family and friend carers followed by friend carers, and/or to thoughts of hurting themselves or someone else, which was most common among family and friend carers, followed by family carers.

Contradicting a previous study [49], friend carers experienced a lower level of support in their caring role in comparison to family carers and family and friend carers. This can partly be explained by the fact that family carers’ and family and friend carers’ respective families receive formal support to a greater extent, which gives the authorities insight into these families and thus increases the possibility of identifying AYCs and offering them support. Only a quarter of the friend carers received formal support, and one out of five stated that school knew about their caring role. This leaves a vast majority of the friend carers with a total lack of support from adults. All three groups seemed to prefer to seek support from a close friend. This result, which clearly shows the importance of having friends to rely on, confirms previous studies [15,16,23]. However, compared to the other groups, fewer friend carers themselves received support from a friend.

In summary, the lack of support, formal as well as informal, was greatest in the friend carer group. This could be due to differences in caring tasks/level of caring, or that young friend carers chose not to tell anyone due to lack of trust in adults, or fear of betraying their friend and potentially making things worse [44].

Friends are important in young people’s lives [15,16,17,23] and even more important in the lives of those young people living with a disability, long-term illness, or mental ill-health [18,19,20,21,22]. The need for friends’ care and support increases during periods of challenges, difficulties [10,11,12], and crises such as the COVID-19 pandemic [13,14].

Since friends play a valuable role for young people with health-related conditions, especially mental ill-health, we need to find ways to optimize the positive aspects and minimize the negative aspects of caring so those adolescents who choose to care for a friend can do so without a serious or long-lasting negative impact on their own situation, including their mental health, and still be able to reach their goals in life. According to a previous study in which young family carers were asked how formal support services can better support them, they tended to request greater support for their care recipient, perhaps indicating that a young carer may feel a less negative impact associated with their caring role if their care recipient is better taken care of [60]. However, further research is needed in order to investigate whether this also applies to young friend carers. The utilization of “young carers projects” in England, which have been evidenced to help reduce stress and encourage young family carers to relax [61], would possibly have a similar effect for young friend carers.

The adult society has a responsibility to ensure that young people’s caring responsibilities do not become too onerous, interfere with their education, or become harmful to young people’s health and development [51] (article 32). Every society should strive to ensure the healthy lives of its participants and promote well-being and lifelong learning opportunities for all [62]. Considering the study results, much remains to be accomplished in fulfilling the UNCRC article 32 and the above-mentioned UNSDGs when it comes to young friend carers and young family and friend carers in the six European countries studied. The perceived impact of AYCs’ caring role on their school situation (school difficulties, school results, been bullied), their self-reported state of health and wellbeing, and that some of them had thoughts about harming themselves or someone else demonstrate an unmet need for support. At the same time, the results show that friend carers experienced the lowest level of support. These are worrying results, since lack of support to those who have a burdensome caring role means an increased risk of developing a vicious circle, with increasing levels of mental ill-health among young friend carers, who, in turn, will need support from other friends, who, in turn, will be at risk of mental ill-health, and so on. In the long run, this equates to an ever-increasing number of young people suffering from mental ill-health.

To achieve a better and more sustainable future for all, this vicious circle must be broken. To reach this goal, we need to focus on prevention and empowerment, starting by recognizing young friend carers, identifying them, listening to their views and needs, and, based on their experiences, co-developing and implementing interventions targeting mental health at an early stage. Such a co-designed intervention was developed and tested in the ME-WE young carers project [50] with the goal of strengthening the resilience of AYCs in transition to adulthood, of promoting their mental health and wellbeing, and of mitigating the negative influence of psychosocial and environmental factors in their lives [63].

In addition, the definition of YCs needs to be expanded to include young friend carers, as in the Carers (Scotland) Act [64], which includes YCs of friends, families, and neighbors. A wider definition of the concept YCs means that young friend carers would be included in current British legislation [65,66]. However, many good practices to support YCs have been implemented within the framework of non-specific legislation [8]. Support in school and/or from child and adolescent mental health service teams, as well as via online platforms for YCs, peer support groups, and summer camps, are good examples of practices that could be used, in their current form or adapted, to support young friend carers.

This study is the first of its kind to provide demographic information on young friend carers at a European level. The large sample of 655 young friend carers and 466 young family and friend carers, spanning six European countries, provides a clear picture of these carers’ characteristics and their caring role. The use of validated instruments (MACA, PANOC, and Kidscreen) provides new knowledge about the positive and negative outcomes of a friend-caring role, as well as its impact on young friend carers’ health, wellbeing, and school situation. It shows that many adolescents play an important role as carers for friends, and that this role is associated with both positive and negative aspects, but with increased risks for their own mental health. It also reveals that compared to family carers, friend carers experienced lower levels of available support. Furthermore, the complex situation of young family and friend carers has been highlighted.

There are, however, some limitations to this study. Recruitment difficulties and obstacles led to deviations from the original recruitment strategy in some countries. Due to the variance between the countries’ sampling strategies and the lack of a known representative sample in each country, it is not possible to present a generalizable country distribution of adolescent young friend carers. Despite these limitations, as this study is the first of its kind, it contributes substantial new knowledge about young friend carers. Further research could usefully explore the extent to which young friend carers receive support from family members.

## 5. Conclusions

Many adolescents in this study provided care to at least one friend with a health-related condition, usually mental ill-health. Some of them also provided care to at least one family member. The number of young friend carers is likely to increase as long as mental ill-health among young people continues to increase. Caring for a friend is often associated with positive, empowering aspects, while at the same time, the caring role can have a negative impact on friend carers’ school situation, general health, and health-related quality of life, and the majority experience a lack of support in their caring role. Thus, there is a need for early preventive measures with a focus on optimizing the positive aspects of young people caring for a friend, and for school staff, youth workers, and other service agencies to actively work to minimize the negative aspects. At a policy level, the definition of young carers needs to include friend carers. Finally, we recommend that researchers working in the field include friend carers in their samples.

## Figures and Tables

**Table 1 healthcare-11-02876-t001:** Respondents in each country who had friends with a health-related condition, and who provided care to that friend.

	Respondents	Had Friends with Health-Related Condition		Provided Care to Friends with Health-Related Condition	
	Total N	N	*(%) **	N	*(%) ***
Sweden	3015	763	*(25.7)*	482	*(66.2)*
UK	724	309	*(43.6)*	169	*(56.3)*
Slovenia	1013	333	*(33.5)*	168	*(55.1)*
Italy	893	210	*(23.8)*	103	*(50.0)*
The Netherlands	630	164	*(26.2)*	74	*(47.4)*
Switzerland	871	276	*(32.1)*	125	*(45.5)*
Total	7146	2055	*(29.2)*	1121	*(56.9)*

* Percentage of all respondents (15–17 years) in each country. ** Percentage of those who had friends with health-related condition (descending order). Note: valid percentages are reported, which do not take account of missing data.

**Table 2 healthcare-11-02876-t002:** Respondents’ gender distributions.

	Female		Male		Transgender		Other	
	N	*(%)*	N	*(%)*	N	*(%)*	N	*(%)*
All respondents (Total N: 7146)	4240	*(60.1)*	2704	*(38.3)*	27	*(0.4)*	87	*(1.2)*
AYCs (Total N: 2099)	1476	*(71.2)*	558	*(26.9)*	15	*(0.7)*	25	*(1.2)*
Family Carers (Total N: 978)	679	*(69.9)*	279	*(28.7)*	6	*(0.6)*	7	*(0.7)*
Friend Carers (Total N: 655)	436	*(67.6)*	195	*(30.2)*	4	*(0.6)*	10	*(1.6)*
Family & Friend Carers (Total N: 466)	361	*(78.8)*	84	*(18.3)*	5	*(1.1)*	8	*(1.7)*

Note: valid percentages are reported, which do not take account of missing data (i.e., those who selected “prefer not to say” in response to gender).

**Table 3 healthcare-11-02876-t003:** Reported health-related conditions among the AYCs’ friends and family members.

	Family Carers’ Family Members		Friend Carers’ Friends		Family & Friend Carers’ Family Members and Friends			
					Family Members		Friends	
	N	*(%)*	N	*(%)*	N	*(%)*	N	*(%)*
Mental ill-health	324	*(33.2)*	429	*(66.2)*	253	*(54.6)*	337	*(72.6)*
Physical disability	465	*(47.6)*	127	*(19.6)*	200	*(43.2)*	96	*(20.7)*
Addiction	70	*(7.2)*	124	*(19.1)*	80	*(17.3)*	98	*(21.1)*
Cognitive impairment	246	*(25.2)*	124	*(19.1)*	127	*(27.4)*	75	*(16.2)*
Other *	255	*(26.1)*	92	*(14.2)*	118	*(25.5)*	50	*(10.8)*

* e.g., long-term physical illness, allergies, fatigue syndrome, aggressive behavior, visual or hearing impairment, and unknown diagnosis. Note: valid percentages are reported, which do not take account of missing data.

**Table 4 healthcare-11-02876-t004:** MACA subscale scores (range 0–6, higher scores indicating greater activity)—mean values.

	Non-Carers (N 5047 *)		Family Carers (N 978 *)		Friend Carers (N 655 *)		Family & Friend Carers (N 466 *)	
	M	*(SD)*	M	*(SD)*	M	*(SD)*	M	*(SD)*
Domestic activity	3.61	*(1.41)*	3.92	*(1.33)*	3.77	*(1.40)*	4.10	*(1.40)*
Household management	2.53	*(1.36)*	2.90	*(1.38)*	2.75	*(1.42)*	3.05	*(1.38)*
Financial/Practical management	0.52	*(1.0)*	0.87	*(1.23)*	0.71	*(1.18)*	0.98	*(1.30)*
Personal care	0.22	*(0.88)*	0.79	*(1.41)*	0.32	*(1.03)*	0.92	*(1.58)*
Emotional care **	0.60	*(1.29)*	2.56	*(1.85)*	2.00	*(1.83)*	3.20	*(1.87)*
Sibling care	1.31	*(1.68)*	1.65	*(1.85)*	1.59	*(1.84)*	2.03	*(2.04)*

* Missing values: Non-carers 165; Family Carers 26; Friend Carers 25; Family & Friend Carers 8. ** Keep the care recipient company/Keep an eye on the care recipient to make sure they are alright/Take the care recipient out.

**Table 5 healthcare-11-02876-t005:** Number of Family Carers, Friend Carers, and Family & Friend Carers scoring below 12 on the PANOC positive scale and above 8 on the negative scale.

	PANOC Positive Score below 12		PANOC Negative Score above 8	
	N	*(%)*	N	*(%)*
Family Carers (N 978)	288	*(29.4)*	162	*(16.6)*
Friend Carers (N 655)	159	*(24.3)*	100	*(15.3)*
Family & Friend Carers (N 466)	155	*(33.3)*	154	*(33.0)*
Total (N 2099)	602	*(28.7)*	416	*(19.8)*

**Table 6 healthcare-11-02876-t006:** Positive and negative impacts of caring, experienced occasionally or frequently by family carers, friend carers, and family and friend carers.

	Family Carers (N 978)		Friend Carers (N 655)		Family & Friend Carers (N 466)	
	N	*(%)*	N	*(%)*	N	*(%)*
Positive impacts:						
Because of caring…						
…I feel I am doing something good	810	*(94.4)*	518	*(95.6)*	406	*(93.3)*
…I feel that I am helping	807	*(94.7)*	519	*(95.8)*	415	*(95.7)*
…I feel closer to my family	738	*(87.7)*	408	*(82.8)*	360	*(84.2)*
…I feel good about myself	700	*(83.3)*	453	*(87.0)*	327	*(76.9)*
…I feel that I am learning useful things	682	*(82.3)*	429	*(84.4)*	367	*(86.4)*
…my parents are proud of the kind of person I am	716	*(86.5)*	412	*(85.3)*	369	*(87.9)*
…I like who I am	653	*(81.0)*	418	*(83.9)*	304	*(72.3)*
…I feel I am better able to cope with problems	602	*(74.6)*	394	*(80.6)*	308	*(72.4)*
…I feel I am useful	718	*(88.1)*	416	*(85.9)*	355	*(83.5)*
I feel good about helping	773	*(93.3)*	473	*(92.8)*	391	*(91.1)*
Negative impacts:						
Because of caring…						
…I have to do things that make me upset	427	*(52.2)*	234	*(48.2)*	252	*(59.8)*
…I feel stressed	517	*(62.0)*	338	*(66.7)*	330	*(76.8)*
…I feel like running away	225	*(28.0)*	160	*(34.0)*	195	*(46.5)*
…I feel very lonely	286	*(35.5)*	164	*(34.5)*	227	*(53.2)*
…I feel I can’t cope	297	*(37.3)*	215	*(45.1)*	217	*(51.7)*
…I can’t stop thinking about what I have to do	415	*(51.2)*	262	*(55.8)*	278	*(65.6)*
…I feel so sad I can hardly stand it	256	*(32.0)*	158	*(33.9)*	199	*(47.4)*
…I don’t think I matter	221	*(27.7)*	142	*(30.3)*	172	*(41.1)*
…life doesn’t seem worth living	157	*(20.3)*	122	*(26.2)*	150	*(36.2)*
…I have trouble staying awake	267	*(34.0)*	168	*(35.7)*	194	*(46.3)*

Note: valid percentages are reported, which do not take account of missing data (i.e., those who did not answer the question or those who selected “not applicable”).

**Table 7 healthcare-11-02876-t007:** Self-reported health conditions.

	Non-Carers (N 5047)		Family Carers (N 978)		Friend Carers (N 655)		Family & Friend Carers (N 466)	
	N	*(%)*	N	*(%)*	N	*(%)*	N	*(%)*
Mental health problems	634	*(13.7)*	243	*(27.0)*	206	*(33.8)*	249	*(56.2)*
Physical disabilities	91	*(2.0)*	54	*(6.0)*	19	*(3.1)*	50	*(11.3)*
Learning difficulties	256	*(5.5)*	91	*(10.1)*	67	*(11.0)*	61	*(13.8)*
Dyslexia	129	*(6.0)*	19	*(8.9)*	21	*(7.0)*	13	*(7.9)*
ADHD	81	*(3.8)*	15	*(7.0)*	17	*(5.7)*	19	*(11.5)*
Asperger or Autism	33	*(1.5)*	6	*(2.8)*	1	*(0.3)*	3	*(1.8)*
Other health-related conditions	362	*(7.8)*	118	*(13.1)*	83	*(13.6)*	63	*(14.2)*

Note. Valid percentages are reported, ignoring missing values.

**Table 8 healthcare-11-02876-t008:** Experienced issues and difficulties due to AYCs’ caring role.

	Family Carers (N 978)		Friend Carers (N 655)		Family & Friend Carers (N 466)	
	N	*(%)*	N	*(%)*	N	*(%)*
Find school difficult because of caring	151	*(16.4)*	50	*(8.0)*	115	*(25.7)*
Caring has a negative impact on school results	149	*(16.2)*	63	*(10.1)*	126	*(28.1)*
Been bullied because of caring	138	*(15.0)*	50	*(8.1)*	107	*(23.8)*
Considered hurting oneself	94	*(10.2)*	71	*(11.5)*	119	*(26.6)*
Considered hurting others	55	*(6.0)*	26	*(4.2)*	39	*(8.7)*
Of those who had considered hurting others:						
The care recipient	30	*(56.6)*	9	*(34.6)*	13	*(35.1)*
Someone else	23	*(43.4)*	17	*(65.4)*	24	*(64.9)*

Note: the valid percentage is presented, ignoring missing values. For the “of those who had considered hurting others” columns, the percentage reflects the valid percentage of the participants who indicated they had considered hurting others.

**Table 9 healthcare-11-02876-t009:** AYCs’ current access to formal and informal support.

	Family Carers (N 978)		Friend Carers (N 655)		Family & Friend Carers (N 466)	
	N	*(%)*	N	*(%)*	N	*(%)*
Formal support in connection to the caring role	290	*(32.4)*	152	*(25.1)*	169	*(38.3)*
The family receive formal support	306	*(34.5)*	61	*(10.1)*	127	*(28.7)*
School knows about the caring role	279	*(31.1)*	125	*(20.9)*	162	*(36.5)*
Employer knows about the caring role	80	*(9.1)*	21	*(3.5)*	40	*(9.2)*
A close friend knows about the caring role	475	*(53.0)*	283	*(47.3)*	305	*(68.7)*

Note: the valid percentage is presented, ignoring missing values.

## Data Availability

The data that support the findings of this study are available on request from the corresponding author. The data are not publicly available due to privacy or ethical restrictions.

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
