# Peer review of "Adolescent Young Carers Who Provide Help and Support to Friends"

_healthcare, 2023, doi:10.3390/healthcare11212876_

Round 1
Reviewer 1 Report
The subject is very interesting and deserves a great deal of interest, as it is a field of study that has not yet been explored to any great extent.
Here are just a few comments.
Introduction:
The introduction is very well written but perhaps a little redundant. Maybe it could be limited a bit.
Materials and Methods:
Was the online questionnaire accessible to the public or was it only accessible via a registration and/or a password?
While I understand the difficulty, I wonder whether the results were influenced by the huge differences in recruitment methods, background and area of residence of the boys and girls, and whether the sample is therefore truly representative (at least in relation to the European countries involved in the project).
Results:
I wonder if there is a difference in outcomes between Young Careers caring for people with mental illness and those caring for people with complex and/or terminal physical illness. And if so, what are they?
Table 1.What is the data order (descending, alphabetical...)?
Discussion
In order to better understand the results, it would also be helpful to know the legal situation in each of the countries participating in the survey with regard to support for carers in general.
It would also be interesting to understand is and how the families of these young careers relate to them and whether they in turn provide support.
Referencees
I suggest eliminating repetitions and typing errors in some bibliographical references (e.g. 42, 48, etc.).
Author Response
Responses to Reviewer 1’s Comments
We would like to thank Reviewer 1 for their highly constructive feedback. Please find our detailed responses to the reviewer’s comments below and the corresponding revisions/corrections highlighted in the re-submitted file.
Introduction
Reviewer’s comment 1: The introduction is very well written but perhaps a little redundant. Maybe it could be limited a bit.
Our response 1: Prior to submitting our manuscript, we had actually reduced and tightened our introduction section. We have carefully considered Reviewer 1’s suggestion but we consider that as the Healthcare Journal is a general healthcare based journal we cannot assume that the general reader has a lot of knowledge of young carers, therefore we consider that the text in the Introduction section is not redundant, but rather helps the reader to gain a concise yet comprehensive overview of the topic, the current state of the art and the rationale for our chosen focus which is an under-explored topic within the existing empirical literature. Our response is also in line with Reviewer’s 2 comment about the importance of setting the context regarding adolescent mental health issues.
Materials and Methods
Reviewer’s comment 2: Was the online questionnaire accessible to the public or was it only accessible via a registration and/or a password?
Our response 2: The details about the accessibility of the questionnaire are clearly outlined on page 4, paragraphs 1–3, lines 154–182. In this re-submitted manuscript we have added that to ensure anonymity, no registration was required for participants to access the survey (page 4, paragraph 5, lines 191–192).
Reviewer’s comment 3: While I understand the difficulty, I wonder whether the results were influenced by the huge differences in recruitment methods, background and area of residence of the boys and girls, and whether the sample is therefore truly representative (at least in relation to the European countries involved in the project).
Our response 3: The question about sample representativity due to differences in recruitment methods, background, and area of residence of the boys and girls – this aspect is discussed as a study limitation on page 12, paragraph 5, lines 453–458.
Results
Reviewer’s comment 4: I wonder if there is a difference in outcomes between Young Careers caring for people with mental illness and those caring for people with complex and/or terminal physical illness. And if so, what are they?
Our response 4: This study has its focus on young Friend Carers. It was a very small percentage of the participating Friend Carers who provided care and support to people with complex and/or terminal physical illnesses. We therefore assess that such a comparison as requested would not give a reliable result. Thus, from our current study findings, we are not able to state if there is a difference in outcomes between YCs caring for people with mental illness and those caring for people with complex and/or terminal physical illness. This would demand a further study of its own with larger sample sizes to determine any differences between different illnesses and conditions.
Reviewer’s comment 5: Table 1. What is the data order (descending, alphabetical...)?
Our response 5: In table 1, data order is descending when it comes to the percentages of respondents (in each country) who provide care to friends with a health-related condition (the column to the right in the table). In this re-submitted manuscript, we have added this information underneath the table (page 5).
Discussion
Reviewer’s comment 6: In order to better understand the results, it would also be helpful to know the legal situation in each of the countries participating in the survey with regard to support for carers in general.
Our response 6: In this re-submitted manuscript, we have added information about the participating countries’ legal situation in the Introduction section, page 3, paragraph 3, lines 131–134. More detailed information is available in the reference [8].
Reviewer’s comment 7: It would also be interesting to understand is and how the families of these young careers relate to them and whether they in turn provide support.
Our response 7: In the Discussion section, we have added a suggestion for further research to investigate the extent to which young Friend Carers receive support from their family members (page 12, lines 458–460).
References
Reviewer’s comment 8: I suggest eliminating repetitions and typing errors in some bibliographical references (e.g. 42, 48, etc.).
Our response 8: The repetitions and typing errors in the references occurred largely because they were the authors’ own references, but this is now rectified in the re-submitted version.
Yours faithfully
Rosita Brolin, PhD and Elizabeth Hanson, PhD.
Reviewer 2 Report
This is a very interesting paper on an important and topical current research area as it addresses the magnitude of adolscent mental health issues as well as the under recognised issue of the care young people provide to their peers (as well as parents/family members/neighbours etc).
There are a few general points I feel could be addressed in this paper, especially the discussion of the findings. The data is specific to the topic of young caregiving, and the authors acknowledge an (unavoidable) limitation of their sampling. However, as the interpretation of the results speaks to the wider issue of adolescent mental health issues, these being under-recognised and the barriers that young people may face in seeking professional support, it would be helpful to contextualise this more clearly. For example, providing statistics about general prevalence of mental ill health in this age group, the influence that social media can have on this group's understanding/perception/knowledge of what mental ill health is etc.
It is important not to imply that only professionally diagnosed mental ill health is of concern or should be acknowledged as problematic or needing support, however, it also seems to be the case that there is a significant concern around over self-diagnosis of problems, especially because of the influence that social media has (e.g. Haltigan et al 2023, https://doi.org/10.1016/j.comppsych.2022.152362)
In the description and discussion of the results it would be helpful to provide some more context for how school situation is defined, and what is meant by emotional care (how extensive is this care, especially compared to the physical care that is more traditionally associated with young carers).
The section on page 7 line 273 onwards is difficult to follow, it would benefit from some rephrasing to improve the clarity.
Author Response
Responses to Reviewer 2’s Comments
We would like to thank Reviewer 2 for their highly constructive feedback. Please find our detailed responses to the reviewer’s comments below and the corresponding revisions/corrections highlighted in the re-submitted file.
Reviewer’s comment 1: The data is specific to the topic of young caregiving, and the authors acknowledge an (unavoidable) limitation of their sampling. However, as the interpretation of the results speaks to the wider issue of adolescent mental health issues, these being under-recognised and the barriers that young people may face in seeking professional support, it would be helpful to contextualise this more clearly. For example, providing statistics about general prevalence of mental ill health in this age group, the influence that social media can have on this group's understanding/perception/knowledge of what mental ill health is etc.
Our response 1: The contextualization of our study is provided in our Introduction section where we reference to the findings of nine different empirical studies that have focused on mental ill-health among children and adolescents – page 2, paragraph 5, lines 77–78.
Reviewer’s comment 2: It is important not to imply that only professionally diagnosed mental ill health is of concern or should be acknowledged as problematic or needing support, however, it also seems to be the case that there is a significant concern around over self-diagnosis of problems, especially because of the influence that social media has (e.g. Haltigan et al 2023, https://doi.org/10.1016/j.comppsych.2022.152362)
Our response 2: In this re-submitted version of the manuscript, we have added that no conclusions can be drawn regarding the proportion of professional diagnoses, as the care recipients’ health-related conditions were self-reported by the YCs (page 10, paragraph 4, lines 334–336). We also added the word “self-reported” in page 11, paragraph 2, line 363. We appreciate the interesting reference on the influence of social media (Haltigan et al., 2023). However, we have chosen not to include this reference in our manuscript, as we consider it not relevant to the focus of our study.
Reviewer’s comment 3: In the description and discussion of the results it would be helpful to provide some more context for how school situation is defined, and what is meant by emotional care (how extensive is this care, especially compared to the physical care that is more traditionally associated with young carers).
Our response 3: The school situation refers to the following three items within the survey: "Find school difficult because of caring" + "Caring has a negative impact on school results" + "Been bullied because of caring" (Table 8). In the re-submitted manuscript, we added this clarification on page 12, paragraph 1, line 417.
The emotional care refers to the following three items: ”Keep the person you care for company e.g. sitting with them, reading to them, talking to them” + ”Keep an eye on the person you care for to make sure they are alright” + ”Take the person you care for out e.g. for a walk or to see friends or relatives”. In this re-submitted manuscript, we have added this clarification underneath table 4 (page 7) where the extent of emotional care and other caring activities are presented.
Reviewer’s comment 4: The section on page 7 line 273 onwards is difficult to follow, it would benefit from some rephrasing to improve the clarity.
Our response 4: The section has now been rephrased to improve the clarity: "A majority of AYCs in all three groups felt good about caring. They reported that caring led them to learn useful things, that caring had positive impacts on their self-view, their relations to families, and their ability to cope with problems. On the other hand, a majority in all three groups felt stressed about caring. They could not stop thinking about their caring responsibilities. Furthermore, about half of the Family and Friend Carers felt very lonely and that they could not cope (table 6)".
Yours faithfully
Rosita Brolin, PhD and Elizabeth Hanson, PhD.
Reviewer 3 Report
I have enjoyed reading this paper which I think provides a very interesting insight into a topic that is often forgotten or ignored, that of young friend carers. I consider that the evidence it provides will add to the body of knowledge around young carers which is in great need of more research.
I have a number of general comments and suggestions for the authors:
Design: because you are explaining the process followed in six different countries I think it would be good to show your recruitment strategies in the form of a table, at the moment one gets confused between countries and what was done in each. This will allow for a way to see it all at a glance. It will also highlight important issues such as for example the fact that the UK recruited via support organizations rather than schools. This table can also explain how the PIS and details were given to students and how and where the questionnaire was filled and how consent was taken (e.g. by email, in class at school) etc.
More detail on the demographic data collected would add to the clarity of the design. Maybe the whole questionnaire could be a supplementary file, or at least the section that are not validated instruments. It will also allow the reader to see what questions were asked for example in terms of the 'condition' of the person they helped care for.
Results: Similarly, I think one of two figures (maybe pie chart style or bars) would help get a quick glance of who was recruited and what the results show. It will help compare among countries but also highlight similarities.
The discussion presents a good level of critique of different areas. I would like it to maybe include a little bit about how the condition of the person the YC helps care is a vital aspect that often gets ignored or not looked into with the right level of detail (e.g., mental ill-health or cognitive impairment are very broad terms and could mean many things). It is a limitation of many similar studies I have been involved with or reviewed recently.
I welcome this research and I consider it relevant and worthy of being published.
I would recommend a detailed edit to spot typos and to shorten some sentences that are difficult to follow. Otherwise the quality of the English is of a good standard.
Author Response
Responses to Reviewer 3’s Comments
We would like to thank Reviewer 3 for their highly constructive feedback. Please find our detailed responses to the reviewer’s comments below and the corresponding revisions/corrections highlighted in the re-submitted file.
Reviewer’s comment 1: Design: because you are explaining the process followed in six different countries I think it would be good to show your recruitment strategies in the form of a table, at the moment one gets confused between countries and what was done in each. This will allow for a way to see it all at a glance. It will also highlight important issues such as for example the fact that the UK recruited via support organizations rather than schools. This table can also explain how the PIS and details were given to students and how and where the questionnaire was filled and how consent was taken (e.g. by email, in class at school) etc.
Our response 1: The recruitment strategy was the same in all countries. However, due to recruitment challenges, the final recruitment differed slightly between countries in terms of where the recruitment took place, as described on page 4, paragraph 3, lines 177–184. In this re-submitted version we have added the reference to the overall ME-WE survey study [56] which provides more detailed information about the recruitment. In page 4, paragraph 7, lines 202–213 it is clearly outlined how the information about the study and the consent forms were given to the adolescents. In this re-submitted manuscript we have added detailed information on how consent was taken (page 4, paragraph 7, line 202 and 207, and page 5, lines 208–211). We consider as we already have 9 tables included in our manuscript that we already have more than sufficient, given that often the optimal number of tables included in an article is approximately 8 tables.
Reviewer’s comment 2: More detail on the demographic data collected would add to the clarity of the design. Maybe the whole questionnaire could be a supplementary file, or at least the section that are not validated instruments. It will also allow the reader to see what questions were asked for example in terms of the 'condition' of the person they helped care for.
Our response 2: In this re-submitted manuscript we have added more detailed information about the collected demographic data (page 3, paragraph 5, lines 145–147) and what questions were asked for in terms of the condition of the YCs’ care recipients (page 3, paragraph 5, and page 4, paragraph 1, lines 151–160).
Reviewer’s comment 3: Results: Similarly, I think one of two figures (maybe pie chart style or bars) would help get a quick glance of who was recruited and what the results show. It will help compare among countries but also highlight similarities.
Our response 3: Who was recruited is reported at the beginning of the results section (page 5, paragraph 3, lines 223–226), followed by all other results presented in the running text and tables. We have considered collating the results, and on a country basis, in one or two figures. However, given the large amount of results, we consider it best to keep the results section in its current form, to maintain clarity.
Reviewer’s comment 4: The discussion presents a good level of critique of different areas. I would like it to maybe include a little bit about how the condition of the person the YC helps care is a vital aspect that often gets ignored or not looked into with the right level of detail (e.g., mental ill-health or cognitive impairment are very broad terms and could mean many things). It is a limitation of many similar studies I have been involved with or reviewed recently.
Our response 4: As written above, in this re-submitted manuscript we have added more detailed information about what questions were asked for in terms of the condition of the YCs’ care recipients (page 3, paragraph 5, and page 4, paragraph 1, lines 151–160). We have also added in the discussion section that it is important to be aware that no conclusions can be drawn regarding the proportion of professional diagnoses, as the care recipients’ health-related conditions were self-reported by the YCs (page 10, paragraph 4, lines 342–344).
Reviewer’s comment 5: Comments on the Quality of English Language
I would recommend a detailed edit to spot typos and to shorten some sentences that are difficult to follow. Otherwise the quality of the English is of a good standard.
Our response 5: Unfortunately, Reviewer 3 doesn’t specify in detail which sentences are difficult to follow. Thus, it is difficult for us to know exactly which sentences are intended. However, in this re-submitted manuscript we have rephrased a paragraph to improve its clarity (page 7, paragraph 3, lines 287–292). The manuscript has been carefully language edited and reviewed several times by three co-authors who are native English speakers, to ensure the quality of the language and grammar.
Yours faithfully
Rosita Brolin, PhD and Elizabeth Hanson, PhD.